# Mesenchymal Stem Cell-Based Therapies in the Post-Acute Neurological COVID Syndrome: Current Landscape and Opportunities

**DOI:** 10.3390/biom14010008

**Published:** 2023-12-20

**Authors:** Lilia Carolina León-Moreno, Edwin Estefan Reza-Zaldívar, Mercedes Azucena Hernández-Sapiéns, Erika Villafaña-Estarrón, Marina García-Martin, Doddy Denise Ojeda-Hernández, Jordi A. Matias-Guiu, Ulises Gomez-Pinedo, Jorge Matias-Guiu, Alejandro Arturo Canales-Aguirre

**Affiliations:** 1Unidad de Evaluación Preclínica, Biotecnología Médica Farmacéutica, CONACYT Centro de Investigación y Asistencia en Tecnología y Diseño del Estado de Jalisco (CIATEJ), Guadalajara 44270, Mexico; lileon_pos@ciatej.edu.mx (L.C.L.-M.); mercedes.azucena@gmail.com (M.A.H.-S.); estarrone.19@gmail.com (E.V.-E.); 2Instituto Tecnológico y de Estudios Superiores de Monterrey, Monterrey 64849, Mexico; edwin.reza@tec.mx; 3Laboratorio de Neurobiología, Instituto de Investigación Sanitaria, Hospital Clínico San Carlos, IdISSC, Universidad Complutense de Madrid, 28040 Madrid, Spain; maring80@ucm.es (M.G.-M.); doddydenise@gmail.com (D.D.O.-H.); jordimatiasguiu@hotmail.com (J.A.M.-G.); u.gomez.pinedo@gmail.com (U.G.-P.); 4Departamento de Neurología, Instituto de Investigación Sanitaria, Hospital Clínico San Carlos, IdISSC, Universidad Complutense de Madrid, 28040 Madrid, Spain

**Keywords:** long COVID, mesenchymal stem cells, exosomes, neurological sequelae

## Abstract

One of the main concerns related to SARS-CoV-2 infection is the symptoms that could be developed by survivors, known as long COVID, a syndrome characterized by persistent symptoms beyond the acute phase of the infection. This syndrome has emerged as a complex and debilitating condition with a diverse range of manifestations affecting multiple organ systems. It is increasingly recognized for affecting the Central Nervous System, in which one of the most prevalent manifestations is cognitive impairment. The search for effective therapeutic interventions has led to growing interest in Mesenchymal Stem Cell (MSC)-based therapies due to their immunomodulatory, anti-inflammatory, and tissue regenerative properties. This review provides a comprehensive analysis of the current understanding and potential applications of MSC-based interventions in the context of post-acute neurological COVID-19 syndrome, exploring the underlying mechanisms by which MSCs exert their effects on neuroinflammation, neuroprotection, and neural tissue repair. Moreover, we discuss the challenges and considerations specific to employing MSC-based therapies, including optimal delivery methods, and functional treatment enhancements.

## 1. Introduction

COVID-19 is caused by the Severe Acute Respiratory Syndrome Coronavirus 2 (SARS-CoV-2), a novel coronavirus that shares more than 96% of the genome sequence with SARS-CoV. This novel coronavirus exhibits clinical symptoms similar to those reported for SARS-CoV and MERS-CoV [1,2]. In June 2023, the World Health Organization (WHO) reported 767,750,853 confirmed cases of COVID-19 and 6,941,095 deaths worldwide. In México, the confirmed cases are more than 7 million and 330 thousand deaths since the first confirmed case on 28 February 2020 [3]. Currently, there is no effective cure for COVID-19 and recovery depends on the immunity of the individuals [4].

Although the mechanisms of Central Nervous System (CNS) infection remain unclear and highly debated, the neurological symptoms of COVID-19 have been described frequently in critically ill patients with comorbidities [5,6]. However, one of the main concerns about these symptoms is that they could be developed by survivors after recovery [5,6] or in patients with mild acute disease, as part of a syndrome defined by the WHO as post-COVID-19 or long COVID [7]. The prevalence, duration, and severity of these symptoms differ among patients [8] and cognitive impairment is one of the most prevalent deficits [9].

Due to the novelty and variability of the neurological long COVID symptoms, finding an effective treatment might be complex because of the multiple factors involved. Cell-based therapies have generated significant interest in alleviating different pathological conditions. These therapies facilitate the regeneration of tissues and organs by replacing damaged cells and, more likely, by stimulating self-repairing processes [10].

Many studies propose Mesenchymal Stem Cell (MSC)-based treatments mainly due to the minor ethical considerations for their use, their relatively easy and accessible sources, isolation processes, and in vitro expansion [11,12]. MSCs are well known for their homing capacity, immunomodulatory effects, and secretion of paracrine factors to repair tissues and exert a functional recovery [13]. Some authors referred to MSCs as medicinal signaling cells [14] because of their paracrine modulation via the secretion of bioactive molecules [14,15] including lipids, proteins, free nucleic acids, and different types of extracellular vesicles (EVs) [16,17]. The exosomes, an EV type, are one of the major active components of the MSC secretome and are considered key players in the molecule transference between MSCs and recipient cells, and trigger most of the therapeutic effect of MSCs themselves [18]. The MSC-derived exosomes have many advantages compared with the administration of their cellular counterparts because nanovesicles exhibit a lower or no risk of mutagenicity, oncogenicity, and low immunogenicity. Also, they have manufacturing advantages such as storage stability and more accessible transportation [19]. For neurological diseases, the main advantage is that exosomes have a higher chance of crossing the blood–brain barrier (BBB) [20].

Several authors propose MSCs and their derivates as promising therapies for the SARS-CoV-2 infection and its sequelae [21,22,23] due to their anti-inflammatory and regenerative properties. These therapies act on neuropathological hallmarks such as neuroinflammation, neuronal death, synaptic failure, impaired neurogenesis, and oxidative stress [24]. MSC-based therapy administration alleviates cognitive impairment in animal models of cerebral small-vessel disease [25], diabetes [26], traumatic brain injury [27], and Alzheimer’s disease [28], which suggests these therapies could be effective in long COVID-19. Nevertheless, there are still several hurdles to overcome for these therapies to become a reality.

This review aims to describe the neurological manifestations of long COVID, addresses the current landscape of the use and safety of MSC and MSC-derived exosomes, and proposes mechanisms that suggest they could be an effective treatment for these neurological complications. Optimal administration routes and formulation enhancements to effectively reach the CNS are also discussed.

## 2. SARS-CoV-2 Neuroinvasiveness and Long COVID

The SARS-CoV-2 infection mechanism involves the spike glycoprotein (S) and the binding with the angiotensin-converting enzyme 2 receptor (ACE2). The protein binding, eased by specific proteases such as transmembrane serine protease 2 (TMPRSS2), makes the virus capable of invading the respiratory and gastrointestinal epithelial cells [29]. Nevertheless, the S can bind to other receptors, such as Neuropilin-1 (NRP1) and dipeptidyl peptidase 4 (DPP4), that facilitate alternative viral entry and transmission in the target cells [30,31]. Although the ACE2 receptor is mainly expressed in pneumocytes, enterocytes, and vascular endothelial cells, this receptor can also be found on glial cells and neurons in the brainstem, the paraventricular nucleus (PVN), nucleus tractus solitarius (NTS), and the rostral ventrolateral medulla making them a potential target for SARS-CoV-2 [32] with a subsequent CNS infection. Despite many published investigations, the mechanisms of viral infection of the CNS remain unclear and highly debated [33].

Two main pathways of virus entry into the CNS have been proposed (Figure 1). In the first instance, sensory or motor nerve endings are infected, along with the subsequent retrograde neuronal transport [1]. Supporting evidence demonstrates that SARS-CoV-2 can penetrate the brain upon intranasal infection, crossing the neural–mucosal interface in the olfactory mucosa, with a further spreading to defined neuroanatomical areas, including the primary respiratory and cardiovascular control center in the medulla oblongata [34]. This neuroinvasiveness pathway is supported by the localization of viral RNA or proteins in sites such as olfactory mucosa and olfactory sensory neurons (OSNs) [35].

Moreover, peripheral nervous system (PNS) components such as neuromuscular junctions might participate in the neuroinvasiveness potential of COVID-19 [36]. In addition, the respiratory tract (the central infection and replication site of SARS-CoV-2) and the digestive system are innervated by plenty of cranial nerves; neurological invasion beyond the olfactory route is probably achieved using these cranial nerves [37].

Another proposed route is the hematogenous route [1]. SARS-CoV-2 can travel through the circulatory system and reach BBB endothelial cells, promoting BBB leakage and the overexpression of coagulation factors, adhesion molecules, and pro-inflammatory cytokines, as well as the formation of multinucleated syncytia and lysis of the infected endothelial cells [37,38].

Regardless of the route of neuroinvasion, the virus can infect multiple cell types that express the ACE2 receptor, such as neurons, astrocytes, microglia, pericytes, endothelial cells, and oligodendrocytes promoting long COVID symptomatology [35,39]. However, CNS damage directly due to the neuroinvasiveness of SARS-CoV-2 does not seem to be the main mechanism in the pathophysiology of neurological symptoms. In this regard, systemic inflammation causes CNS inflammation through chemokines and other mechanisms, probably causing oligodendrocyte dysfunction, neurogenesis failure, axonal damage, and astrocyte changes [40]. Other mechanisms involved are the development of autoimmunity phenomena, thrombotic and microvascular damage, and reactivation of other infections. Additionally, hypoxic and metabolic issues are also present, especially in cases with severe acute COVID-19.

The broad range of neurological manifestations in acute and chronic stages includes mild and non-life-threatening manifestations such as anosmia, ageusia, headaches, dizziness [41], and myalgias [42], as well as severe manifestations, such as febrile seizures, cognitive impairment, convulsions, peripheral neuropathies, encephalitis [43,44,45], brain edema, and partial neurodegeneration [46]. These manifestations are more frequently presented in critically ill patients with comorbidities. However, they could be developed after their recovery from the primary infection [5,6].

According to clinical findings, about 10–30% of patients experience the symptom persistence of acute COVID-19 or experience new symptomatology after COVID-19 resolution [47]. These symptoms are part of a syndrome defined in 2021 by the WHO as a post-COVID-19 condition, which occurs in individuals with probably or confirmed SARS-CoV-2 infection in the past 3 months as a new onset, following recovery or persists from the initial illness [7]. Other terms to describe it are long COVID syndrome, persistent post-COVID, or post-acute COVID-19 [6]. In this review, we will refer to it as long COVID.

Although SARS-CoV-2 infection is less severe apparently causing milder symptoms, fewer hospitalization rates, and minor adverse outcomes, and its mortality rate is lower due to vaccines, something to consider is that vaccinated individuals could still be infected with less severe symptomatology [4]. Moreover, long COVID is not only presented in patients with a severe infection that led to hospitalization or intensive care but also in patients that did not require hospitalization [48].

In some cases, long COVID includes a broad range of manifestations in the CNS. After 6 months of the COVID-19 infection, significant rates of neurological and neuropsychiatric symptoms have been identified in up to one third of the recovered patients [49]. These neurological manifestations are: difficulty thinking or concentrating (also mentioned as “brain fog”), changes in smell and taste, sleep problems, depression, and headaches [6,9], cognitive impairment, mood changes, anxiety, insomnia, headache, anosmia, and ageusia [50]. Also, 12 months after COVID-19 infection there is an increased risk of stroke, disorders in cognition and memory, sensory, movement, mental health, musculoskeletal and PNS impairments, and other manifestations including Guillain–Barré syndrome, encephalitis, encephalopathy, and extrapyramidal and episodic disorders [48]. However, cognitive impairment is one of the most prevalent symptoms [9,51].

Despite many published investigations, the exact pathologic basis for these neurologic symptoms remains unknown [8,52]. However, the persistent symptomatology of long COVID could have multiple origins due to the different treatment protocols and intensity of infection of every patient, co-morbidities, or high-risk factors [8,53,54]. The evidence shows that severely ill patients tend to have a high concentration of pro-inflammatory cytokines, such as interleukin IL-6, interleukin-1β, CXCL10, IL-2R, TNF-α, and IFN-γ, is associated with cytokine storms (CSs) [55,56]. CSs appear to be aggravated by IL-6, resulting in the chemotaxis of neutrophils and lymphocyte exhaustion [55]. Unfortunately, the high level of cytokines also indicates a poor prognosis for COVID-19.

The CNS is susceptible to CSs, which can damage neurons, astrocytes, microglia, pericytes, endothelial cells, and oligodendrocytes, and promote the disruption of the BBB, which in turn could lead to immune cell infiltration, promoting further inflammatory response enhancement, including the overproduction of pro-inflammatory cytokines, astrocyte/microglial activation neuroinflammation, and finally neurodegeneration [57,58]. Neuroimaging studies have revealed important insights, confirming that cognitive dysfunction in patients with long COVID is associated with structural and functional brain changes [59,60,61].

The neurological symptoms of long COVID are a growing problem and a call for attention for the healthcare system, because they require planning and the development of effective treatments [48,62], a challenging task given the pathophysiology and the interaction of numerous factors [63]. In this review, we propose MSC-based therapies as a promising approach to prevent and alleviate these sequelae.

## 3. Current Landscape of MSC and MSC-Derived Exosomes in Long COVID

Since the beginning of the COVID-19 pandemic, several authors suggested MSC and their derivates, including conditioned media, extracellular vesicles, and exosomes, as a promising therapy for the SARS-CoV2 infection [19,64]. The hypothesis was that these therapies could induce an immunomodulatory response against CSs along with improved regeneration of damaged tissue and improved lung function, mainly through the secretion of bioactive molecules [14,65], as well as because of the positive results obtained from preclinical models of acute respiratory distress syndrome (ARDS) [66,67], influenza [68,69], and other respiratory virus infections [70] in which MSC or its derivates improved animal outcomes and survival rates, mitigated pulmonary and systemic inflammation, and evidenced safety [23,70].

After the release of the first study of a successfully treated COVID-19 patient with MSC in China [71], several pilot trials and case reports appeared in which MSC or MSC-derived treatments were administered alone or in addition to the COVID-19 standard treatment [72]. The preliminary results were favorable in critically ill patients with poor prognoses, showing that these therapies could restore oxygenation levels and lung function, and downregulate CSs (Table 1).

To date, more than 100 clinical trials registered on the clinicaltrials.gov website are exploring the effects of MSC and their derivates in COVID-19. The results of the finalized trials are published in the PubMed database and describe the administration of MSC from different origins, as well as MSC-derived exosomes, EVs, and their secretome [77,78,79,80]. The main endpoint of those trials was to demonstrate safety and tolerability. All trials concluded that these therapies are completely safe, and no severe adverse events were observed. Another secondary endpoint was the efficacy of MSC-based therapies, based on the survival rate, clinical and laboratory improvements, such as the control of CSs. However, these results were not satisfactory [77,80,81,82,83]. While MSC and MSC-derived therapy administration demonstrate beneficial effects in the trials that recruited severe or critically ill patients, the results of the effect of those therapies in patients with mild-moderate symptoms or with low clinical risk were inconclusive. This was mainly explained due to the small number of subjects enrolled in those trials. Therefore, additional clinical investigation is recommended [84,85].

To date, the total evidence indicates that MSC-based therapies improve the symptoms of critically ill or severe patients more significantly than in mild cases [86,87]. This is similar to what is observed with other treatments, like corticoids [62]. However, as the pandemic has evolved, the main concerns are now around the long-term post-COVID sequelae. As described before, the neurological pathophysiology of long COVID is independent of the acute phase, could be present regardless of initial disease severity [88], and could appear in mild and moderate COVID-19 patients [89]. On the way to developing an MSC-based therapy for these sequelae, it is important to understand if those therapies will help patients with different symptom severity or not. On the other hand, the results reported to date and most of the ongoing trials reported in clinicaltrials.org are focused mainly on the effects of these therapies on lung damage, pneumonia, and the control of CSs. Clinical trials exploring the effectiveness of MSC-based therapies on the neurological complications of long COVID are urgently needed.

However, these types of therapies have been already employed in the research of neurological and neurodegenerative disorders [90,91,92] and currently, most clinical trials administering MSC-based therapy are intended to treat neurological conditions [93]. The administration of MSCs provides significant neuroprotection, and induces significant neuro-regeneration and improvement of functional outcomes in preclinical studies [94,95,96,97,98,99,100,101,102]. Although the MSCs have a very low rate of differentiation, often without a clear distinction of neural functionality, and do not survive long-term after brain implantation [103,104], their paracrine activity promotes most of their therapeutic effects, involving processes such as immunomodulation, and induction of neuroplastic events such as neurogenesis, neuroprotection, synaptogenesis, and angiogenesis [94,105,106].

Several experimental stroke studies show that MSCs contribute inflammatory resolution through multiple mechanisms including the secretion of soluble factors and/or cell–cell contact [105,107]. In this line, MSCs influence the polarization state of microglia promoting the M2-like phenotype probably by interference with thrombin mediated M1 polarization [108]. MSCs also attenuate the expression of TNF-α, apparently by the elevated expression of IL-6 and VEGF, providing neuroprotection and alleviating the expression of proinflammatory cytokines [109]. Moreover, the TGF-β secreted by MSCs also plays a critical role in shifting microglia with an M1-like phenotype towards the M2-like phenotype [110]. Although many growth factors and cytokines are involved in the immunomodulatory activity of MSCs including IL-10, IL-23/IL-17, MMP2, TGF-β1, HGF, NGF, pGe2, TLR-4, and RAGE [111,112], the underlying anti-inflammatory mechanisms have yet to be validated.

In addition to the immunomodulatory activity, MSCs also enhance functional recovery by endogenous neurogenesis and the up-regulation of synaptic plasticity linked to releasing neurotrophic factors such as FGF, VEGF, NGF, NT-3, SDF-1, and BDNF [105,113,114]. Increasing level of these neurotrophic factors activate several pathways promoting the survival, proliferation, and differentiation of neural precursor cells [114]. The co-culture of MSC with neural precursor cells increases the expression of proliferative markers as well as progenitors and neuronal markers. Furthermore, MSCs increase the expression of beta catenin and Ngn1, indicating that MSCs have a role in the commitment of the neuronal fate of neural precursor cells by increasing the Wnt signaling pathway [115]. Additionally, MSCs have the ability to induce axonal growth [116,117]. In a recent study, the injection of MSCs overexpressing FGF-21 corrected the abnormal TBI-induced dendritic morphology of immature newborn neurons [118].

Although the effectiveness of MSC therapy regarding genuine cell replacement remains limited considering the very limited MSC transdifferentiation, several studies support that the neuroprotective potential of MSCs relies on their secretome [94,105,106], a set of secreted bioactive molecules which are either dissolved in the cell medium or encapsulated within EVs [119]. This MSC-derived secretome stimulates endogenous self-repair processes, such as the proliferation and differentiation of neural stem cells, as well as neuron maturation and survival, resulting in positive outcomes [92,120,121].

In this line, preclinical studies support the use of MSC-derived exosomes in neural regeneration approaches [122,123,124,125]. Proteomic analysis of MSC-derived exosomes resulted in the identification of more than 900 proteins [126,127,128], including filamin-A, BDNF, vinculin, NGF, FGF, neuropilin-1, VEGF, neuroplastin, glia-derived nexin, DPYSL2, flotillin-1, ephrins, drebrin, neprilysin, teneurin-4, and stathmin, which induce neurogenesis and myelin formation, promote neurite outgrowth and branching, stimulate axonal growth and regeneration, and provide neuroprotection to injured neurons [120,129]. Moreover, their broad cytokine repertoire can efficiently inhibit the effector function of the inflammatory M1-like phenotype and induced the generation of the anti-inflammatory M2-like phenotype in microglial cells, as well as contribute to ameliorating cognitive alterations associated with inflammatory states [130,131]. Moreover, MSC-derived exosomes exhibit properties and cell functions without the controversial long-term fate of MSCs [132]. For instance, the MSC-derived exosomes exhibit a lower or no risk of mutagenicity, oncogenicity, and very low immunogenicity. For CNS targeting approaches, the main advantage of exosomes is their higher capacity to cross the BBB [20]. In addition, they have manufacturing advantages such as storage stability and more accessible transportation [19]. Therefore, the use of stem cell-derived exosomes has also been proposed as a treatment option for long COVID.

On the way to developing and optimizing a cell-based therapy for long COVID, several parameters need to be controlled [87,93]. Among the most important determinants of the success of MSC-based therapies in neuropathies is deciding on the optimal delivery route to ensure that the treatment will reach the CNS [133]. In this line, some key factors that will determine the efficacy of the delivery are delivery to the olfactory area as opposed to the respiratory region, the dose volume, the retention time at the nasal mucosal surface, penetration of nasal epithelia, and a reduction of compound metabolism in the nasal cavity. In this context, using nanoparticles, penetration enhancers, and matrices like hydrogels could improve the delivery to the brain via the nose-to-brain route.

## 4. Administration Routes of MSC and MSC-Derived Exosomes for Neurological Diseases

One of the biggest challenges of developing therapies for the nervous system is the delivery of treatment due to natural barriers. In cell-based therapies, we must customize the delivery route according to the targeted disease and the patient’s circumstances [134]. If the MSCs need to be in the injury site to exert their effects, the optimized and accurate delivery of cells to the injured tissue is a major determinant of overall success [16,133,135]. However, when the distal effects of the MSC-derived secretome can relieve the pathology of the disease, it may not be necessary for the cells to be located at the injury site, and we can use systemic routes [136]. It is important to highlight that delivery route efficacy can vary depending on the disease and target tissue, thus, the amount of MSC derivates necessary in the parenchyma to achieve the expected biological effect must be considered [135].

Different delivery methods have been used before in clinical trials to reach the nervous system. The most prevalent route for MSCs in clinical trials for COVID-19 is the intravenous route (IV) [19]. However, these trials were not focused on a specific pathology but on controlling the severity of the disease. An analysis of the MSC clinical trials from 2004 to 2018 showed that the most used administration routes for neurological disorders were IV and intrathecal (IT), followed by intra-muscular (IM), intra-arterial (IA), and intra-osseous (IO), probably because these routes matched with the targeted tissue [93]. Another work considering 71 clinical trials that used MSC for neurodegenerative diseases found that the most used route was IT, followed by IV. Other methods included administration into the injury site by surgery [137].

Systemic IV injection is a widely used, non-invasive delivery method mainly because of its relative ease of administration. However, this route exhibits some disadvantages, for instance, following IV administration, most MSCs become trapped in the lungs due their relatively large diameter [138,139]. Although the accumulation prolongs the presence of MSCs in the tissue and may provide the delivery of therapeutic agents that could remove chronic complications such as in COVID-19, the long-lasting effects of the MSCs remain to be determined to eliminate potential health risks [16]. For example, physical obstruction with MSCs may represent a microthrombus risk [140].

Different studies described that the positive effects of MSCs in the brain are independent of the existence of a vast number of cells in the injury area or their integration into the tissue [141,142,143]. The IV route has proved effective in pre-clinical and clinical MSC trials for neurological diseases, even when it does not specifically target the nervous system [133], probably due to the secretion of paracrine substances to the systemic circulation activates endogenous repair mechanisms in the brain [142].

Recently, a report proposed the addition of hemocompatibility as a critical characteristic and safety criterion for MSC therapies intended for intravascular use before clinical use [144] because of the thrombogenic effect observed with the IV administration of MSCs and exosomes. MSC Procoagulant Activity (PCA) is generated because of the high Tissue Factor (TF) and low Tissue Factor Pathway Inhibitor (TFPI) expression [145]. Chance et al., (2019) showed that extracellular vesicles derived from AD and BM-MSC are also functionally thrombogenic and express the procoagulant molecules TF and phosphatidylserine on their surfaces [146]. Importantly for the manufacturing process, PCA increases with ex vivo expansion and cryopreservation [145], tissue source, and dose [145,146]. Understanding the interaction between MSC and coagulation is crucial to minimize a patient’s risk and safely delivering MSC therapy, especially when administering to COVID-19 patients [147].

Another systemic and less invasive delivery method is IA, a widespread MSC transplantation method after cerebral infarction. Some studies have shown that significantly larger numbers of MSCs migrate to peripheral tissues compared to the IV route [140,143] probably because the researchers selected the arteries that feed the targeted tissue, implanting MSCs the most naturally and allowing the greatest possible concentration in the tissue [148]. In neurological disorders, the IA administration of both MSCs and their EVs into the internal carotid artery seems to be an effective method for cell injection into the brain [149]. Data shows that some transplanted MSCs attach to the walls of the vessels in the brain temporarily after the injection. Then, these cells can either return to the bloodstream or penetrate through the blood–brain barrier (BBB) to undergo homing in the perivascular niche or penetrate deeper into the parenchyma [150]. Using the IA delivery method, MSCs achieved regeneration through both the paracrine secretion capacity and its integration into the host tissue [140].

BBB integrity could play an essential role in therapy distribution through this route. One study failed to identify MSCs in the brain using an animal model with an intact BBB [135]. This is a critical issue to consider as the patients that will use the MSC-derived therapy for the neurological sequelae of long COVID experienced mild or low symptoms that do not disrupt their BBBs. However, transmigration across the BBB may not be necessary due to the paracrine effects of the cells inside the blood vessels [150]. Another important consideration is that the IA route poses the risks of microembolization and bubble formation, increasing the risk of cerebral lesions [143].

The IT route is the second most popular delivery method for neurological disorders since it administers cells directly into the cerebrospinal fluid (CSF), covering the entire neuraxis. It infuses the MSCs into the subarachnoid space and allows for higher concentrations of cells to migrate to the lesion site compared with systemic administration [151]. This safe route does not require brain surgery, avoiding serious complications such as needle tract injury, infection, and bleeding and lowering the medical cost and psychological burden of surgical procedures [152]. To date, the IT administration of MSCs has shown efficacy for various neurological conditions, including multiple sclerosis, autism, traumatic brain injury, and more, without serious adverse effects, infections, clinical rejection, or tumors [153].

Direct intracerebral transplantation has the inherent advantage of bypassing the BBB to deliver MSCs into the brain [154]. This may allow more direct MSC homing but also results in poor cell distribution in the injured brain and almost no distribution in the peripheral nerves. Moreover, this administration method is highly invasive due to the necessary craniotomy and could result in bleeding, seizures, and other complications that hurdle its clinical translation [151]. These complications can lead to further tissue damage, inflammation, and rejection [155]. Additionally, the localized injection does not allow the cells to exert secondary signaling events in different organs and limit their repertoire of therapeutic effects [154]. Significant risks unique to direct intracerebral delivery have been reported, including differentiation into problematic tissue or ectopic tissue formation [156]. Other considerations include that the cell suspension volume and dose should be as small as possible to alleviate the compression effect [157] without losing sight that the cell survival rate after implantation is 5 to 10% of the injected cells [158].

The Intranasal (IN) route of stem cell administration is an opportunity for the efficient delivery of stem cells directly to the brain parenchyma because it is a non-invasive, rapid absorption method that allows for the penetration of BBB [159]. It uses the olfactory and respiratory pathways and the nasal vasculature to enter the brain tissue [160]. Three transport steps are necessary for delivery to the nervous system after IN administration: across epithelial barriers, from the nasal mucosa to brain entry sites, and from those sites to the parenchyma [161]. IN-administered stem cells appear to cross the olfactory epithelium and enter the subarachnoid space crossing the cribriform plate via the fila olfactoria [162]. To date, only one clinical trial has proved the feasibility and safety of intranasally administered MSCs [163]. More studies are needed to better understand this administration route.

Although the correct administration route is critical to reach the CNS, there are other approaches to guarantee the distribution of MSCs and MSC-derived exosomes in a determined zone. In this sense, diverse strategies, including formulation enhancement have been designed to achieve this goal.

## 5. The Use of Biomaterials as Formulation Enhancement to Optimize Therapeutic Effectiveness

Even though cell therapy holds great promise for restoring damaged neural networks, target tissue delivery remains to be optimized to grant a therapeutic effect [164]. In this line, several approaches for the delivery of cells and their derivates to the brain have been explored. For instance, biomaterials have been integrated into cell-based therapy with MSC or exosomes (Figure 2) to achieve a targeted and controlled release, as well as increase the half-life of these therapies and aid in the development of new tissue-specific regenerative strategies [165].

Among the biomaterials, hydrogels are the most used since they can reduce the effect of mechanical tension during the administration of MSC or EVs, capture large amounts of water and do not cause an immunoreaction [166]. Hydrogels are cross-linked networks of three-dimensional hydrophilic polymers that can mimic the structure of the extracellular matrix, as well as have tunable physical attributes that can be used to synchronize matrix degradation rates to distribute trapped exosomes [167], and can also be designed to achieve stimuli-sensitive gelling, which facilitates their administration and allows them to fill cavities [168].

Common polymers used to produce hydrogels come from two sources (Figure 2); natural such as chitin, collagen, hyaluronic acid, chitosan, alginate, cellulose, keratin, and fibrin, among others, or synthetic such as polyethylene glycol, poly(hydroxyethyl) methacrylate, polytetrafluoroethylene, and polylactic acid [167]. Interestingly, biomaterials have evolved from “bioinert” materials to sophisticated substrates, with the ability to instruct cells and adjust their behavior [169]. Chemical groups of different characters have been selected to transmit a wide range of properties to the biomaterial surface, which are more similar to the native microenvironments of MSCs. For example, carboxylic acid groups are a predominant chemical functionality of cartilage matrices, which are rich in glycosaminoglycans; negatively charged phosphate groups are present in the mineral phase of hard tissues, such as bone; and hydrophobic residues may be associated with adipose tissues, since adipocytes are rich in lipids and secrete them into their extracellular milieu [170,171]. The matrix to provide cells with an environmental structure capable of supporting cell viability, proliferation, and nutrition [172].

On the other hand, combining exosomes and biomaterials could offer advantages, such as the inherent failure of immune system stimulation and transplant rejection [167]. Furthermore, the cells’ behavior can be modulated to modify the exosomes’ capabilities for triggering specific signaling pathways [169]. The strategies to modify exosomes include genetic engineering and cell chemical modification [173]. With these techniques, active molecules can be encapsulated or enriched within exosomes. Unlike synthetic nanoparticles that can only be loaded with drugs during their synthesis, exosomes can be loaded with molecules before or after they are secreted [174]. Also, exosome surface modification is performed to enhance the delivery of these vesicles to specific sites. This modification involves the fusion of the vesicle membrane with a variety of surface adhesion proteins such as integrins, CD11b and CD18 receptors, and antibodies among others to provide better CNS targeting [175]. For instance, Álvarez-Erviti et al. 2011 developed modified exosomes by fusing the neuron-specific rabies viral glycoprotein peptide to the extra exosomal N-terminus of Lamp2b, an abundant exosome membrane protein, to allow exosomes to enter the brain efficiently [174,176].

Exosome modification must be strictly controlled to avoid exosome disruption and aggregation and it must be considered that the introduction of a targeting moiety could reduce exosome multifunctionality. Several studies suggested that exosome engineering enhances their regenerative and anti-inflammatory potential.

Thus, engineered cells and exosomes can be an integrative aspect in designing therapeutic strategies for tissue repair, maintaining cellular homeostasis, or impairing long COVID symptoms.

## 6. Conclusions

Regardless of the different routes of neuroinvasion and the relevance of neuroinvasion in the pathophysiology, it has been demonstrated that SARS-CoV-2 promotes several neurological changes and complications that can compromise the functional dependency of patients. Due to the lack of a specific treatment for these types of complications, the development of strategies involving the use of MSC and their derivatives has been proposed based on results from previous studies in different types of neurological and neurodegenerative models. The proposal for the use of this type of therapy is that it could ameliorate cognitive deterioration and slow down the potential degenerative processes underlying long COVID. However, more studies are needed to determine the efficacy of MSC-based therapies to treat long COVID complications.

## Figures and Tables

**Figure 1 biomolecules-14-00008-f001:**
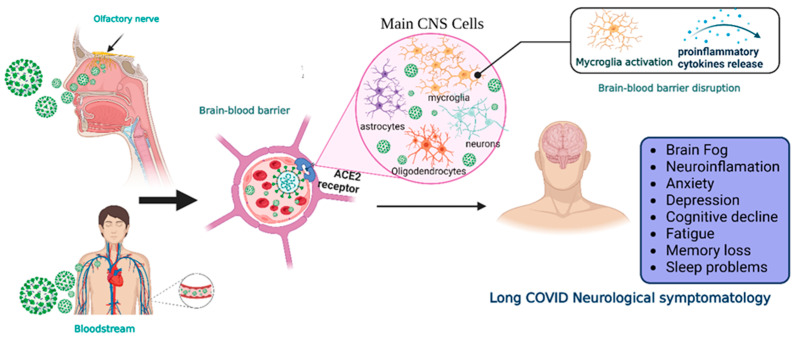
Neuroinvasiveness routes of SARS-CoV-2. Once the virus is in the respiratory system, it can reach the central nervous system by two main mechanisms.SARS-CoV-2 makes contact with the olfactory mucosa, reaching the olfactory nerves, and by transport through the nerve endings, it can travel and spread to the CNS. The other pathway is the hematogenous route, where the virus can reach the brain-blood barrier and, by transcytosis, infect the neuroepithelia and then the cells of the CNS. Viral RNA is present in neurons, astrocytes, oligodendrocytes, and endothelial cells. After the prolongated symptoms, people with long-COVID-19 develop neurologic sequalae. CNS = central nervous system, ACE2 = angiotensin-converting enzyme 2 receptor. Created with BioRender.com.

**Figure 2 biomolecules-14-00008-f002:**
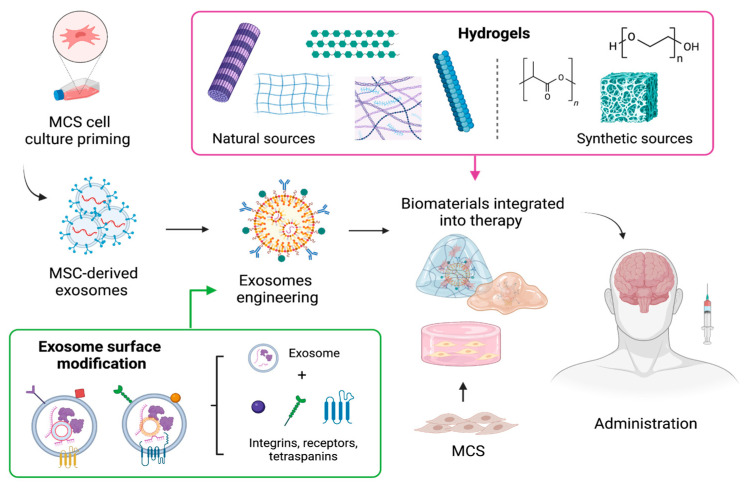
Formulation approaches for MSCs and their cell-derived exosomes using biomaterials to improve MSC-based delivery, absorption, and therapeutic potential. Created with BioRender.com.

**Table 1 biomolecules-14-00008-t001:** Pilot studies and case reports using MSC and MSC-derived exosomes for COVID-19 treatment.

Study Type	Patients Included	MSC or Exosomes	MSC Origin	Treatment Description	Admnistration Route	Dose	Results	Reference
Case report	1	MSC	Umbilical cord (UC)	MSC transplant combined with standard therapeutics for a critically ill COVID-19 patient with severe lung inflammation	Intravenous	3 doses: 5 × 10^7^ cells	No obvious side effects after the MSC transplant Reduced levels of: CRP, ALT, AST, and D-dimer Normal levels of white blood cell, neutrophil, and lymphocyte counts	[71]
Case report	1	MSC	Umbilical cord	UC-MSC therapy + recommended treatment	Intratracheal and Intravenous	2 doses: 0.7 × 10^6^ cells/kg intravenous and 0.3 × 10^6^ cells/kg intratracheal	No adverse effects due to the treatment Improvement in acidosis, electrolyte imbalance and hypoxemia, normal CRP levels, regression in lung infiltrations	[73]
Pilot study	7: MSC transplant 3: Placebo	MSC	Non-Specified	Clinical grade MSCs suspended in 100 mL of normal saline or 100 mL normal saline	Intravenous	1 × 10^6^ cells/kg	Safety: MSC proved to be safe Efficacy: CRP level and biochemical indicators of liver and miocardium damage decreased, improvement of oxygen saturation and lymphopenia	[74]
Pilot study	29: Control 12: MSC treatment	MSC	Umbilical cord	Standard treatment or Standard treatment + UC-MSC MSCs in 100 mL of normal saline	Intravenous	2 × 10^6^ cells/kg	Safety: None MSC treated had adverse reactions Efficacy: CRP and IL-6 levels decreased, improvement of oxygenation index, normal lymphocyte counts, reduced lung inflammation	[72]
Proof of concept study	13	MSC	Adipose tissue (AT)	AT-MSCs, in a medium containing AB serum + 10% dimethylsulfoxide (DMSO) or resuspended in Ringer’s lactate + 1% albumin	Intravenous	1st dose—1 × 10^6^ cells/kg 2nd dose—48–96 h later if considered appropriate	Safety: No adverse effects Efficacy: radiological improvement, extubation, decrease in CRP levels, D-dimer, ferritin and LDH, increase in lymphocyte counts	[75]
Pilot trial	7	Exosomes	Umbilical cord	Exosomes derived from MSCs diluted to 5 mL with 0.9% sodium chloride	Nebulized	7.66e + 0.8 to 7.00e + 0.7 particles/mL	Safety: no acute or secondary allergic reactions and no adverse events Efficacy: efficacy is observed for patients of mild cases of COVID-19 pneumonia	[76]

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
