# Peer review of "Mesenchymal Stem Cell-Based Therapies in the Post-Acute Neurological COVID Syndrome: Current Landscape and Opportunities"

_biomolecules, 2023, doi:10.3390/biom14010008_

Round 1

Reviewer 1 Report

Comments and Suggestions for Authors

The reviewed manuscript presentsthe proposal for use the stem cells of therapy ameliorate the cognitive deterioration and slow down the potential neurodegenerative processes underlying the long COVID. The aim of the work and reference used by the authors are adequate. Therefore, in my opinion the paper is worth to be published in Biomoleculs. Some additional data should be completed:

- line 102; make the font larger

- line 10 Fig 2; illegible figure

Author Response

Thank you for your observations and opinion!

We have addressed all the comments in the manuscript

Reviewer 2 Report

Comments and Suggestions for Authors

English should be checked.

Acronyms should be included the first time they are used, see BBB and IT.

Paragraph 2. SARS-COV-2 neuroinvasiveness and Long COVID

Please avoid unnecessary repetition, some concepts are repeated twice.

Paragraph 5. Formulation enhancement

Choose another heading, you are dealing with biomaterials enhanced with cells or cell derivatives

Table 1. You should provide a column where MSC treatment are clearly separated from exosomes.

Line 219. Table 2 is erroneously referenced; I suppose that it should be table 1.

Comments on the Quality of English Language

Minor editing required.

Author Response

Thank you for your comments, please find the point-by-point responses in italics:

  • English should be checked.

We will check it again before resubmission.

  • Acronyms should be included the first time they are used, see BBB and IT. 

BBB is included in line 67. IT is included in line 324.

  • Paragraph 2. SARS-COV-2 neuroinvasiveness and Long COVID

Please avoid unnecessary repetition, some concepts are repeated twice. 

Thank you for the observation, we will rephrase the paragraph and address the repeated concepts.

  • Paragraph 5. Formulation enhancement

Choose another heading, you are dealing with biomaterials enhanced with cells or cell derivatives

We've changed the heading of the paragraph to better illustrate what is addressed in it.

  • Table 1. You should provide a column where MSC treatment are clearly separated from exosomes.

We added a column to address this comment

  • Line 219. Table 2 is erroneously referenced; I suppose that it should be table 1.

The reference to table two should have been removed since we decided not to include it. It will be deleted from the text